# The Interplay of Adipokines and Pancreatic Beta Cells in Metabolic Regulation and Diabetes

**DOI:** 10.3390/biomedicines11092589

**Published:** 2023-09-21

**Authors:** Joon Kim, Chang-Myung Oh, Hyeongseok Kim

**Affiliations:** 1Department of Biomedical Science and Engineering, Gwangju Institute of Science and Technology, Gwangju 61005, Republic of Korea; joon08167@gm.gist.ac.kr; 2Department of Biochemistry, College of Medicine, Chungnam National University, Daejeon 35105, Republic of Korea; 3Department of Medical Science, College of Medicine, Chungnam National University, Daejeon 35105, Republic of Korea

**Keywords:** adipokines, adipose tissue, pancreatic beta cells, metabolic diseases, diabetes mellitus

## Abstract

The interplay between adipokines and pancreatic beta cells, often referred to as the adipo-insular axis, plays a crucial role in regulating metabolic homeostasis. Adipokines are signaling molecules secreted by adipocytes that have profound effects on several physiological processes. Adipokines such as adiponectin, leptin, resistin, and visfatin influence the function of pancreatic beta cells. The reciprocal communication between adipocytes and beta cells is remarkable. Insulin secreted by beta cells affects adipose tissue metabolism, influencing lipid storage and lipolysis. Conversely, adipokines released from adipocytes can influence beta cell function and survival. Chronic obesity and insulin resistance can lead to the release of excess fatty acids and inflammatory molecules from the adipose tissue, contributing to beta cell dysfunction and apoptosis, which are key factors in developing type 2 diabetes. Understanding the complex interplay of the adipo-insular axis provides insights into the mechanisms underlying metabolic regulation and pathogenesis of metabolic disorders. By elucidating the molecular mediators involved in this interaction, new therapeutic targets and strategies may emerge to reduce the risk and progression of diseases, such as type 2 diabetes and its associated complications. This review summarizes the interactions between adipokines and pancreatic beta cells, and their roles in the pathogenesis of diabetes and metabolic diseases.

## 1. Introduction

Adipokines are a group of signaling proteins produced by the adipose tissue. They play key roles in regulating various physiological processes including energy metabolism, inflammation, insulin sensitivity, and appetite control [1]. Disrupted adipokine production and secretion are strongly associated with metabolic disorders such as obesity, type 2 diabetes, and cardiovascular diseases. Adipokines influence insulin sensitivity, and their imbalance can directly contribute to the development and progression of metabolic disorders [2]. Adipokines comprise a diverse group of signaling molecules, including hormones, cytokines, chemokines, and growth factors. Some well-known adipokines include leptin, adiponectin, resistin, visfatin, and tumor necrosis factor-alpha (TNF-α). Each adipokine has unique biochemical characteristics and performs specific physiological functions that contribute to the maintenance of metabolic homeostasis.

The pancreatic beta cell plays a critical role in maintaining glucose homeostasis. Beta cells synthesize, store, and release insulin, the hormone that is primarily responsible for lowering blood glucose levels. The dysfunction or loss of beta cells, which can arise due to various factors, including autoimmune attacks, inflammation, or metabolic stress, leads to inadequate insulin production and secretion. This results in elevated blood glucose levels, which is a primary characteristic of diabetes. In type 1 diabetes mellitus (T1DM), an autoimmune process targets and destroys beta cells, leading to an absolute insulin deficiency [3]. In contrast, type 2 diabetes mellitus (T2DM) is caused by increased insulin resistance in peripheral tissues. The body attempts to compensate for this insulin resistance by augmenting insulin secretion. However, beta cells may fail to sustain this compensatory response owing to functional impairment, and loss of beta cells results in a relative insulin deficiency [4]. Both T1DM and T2DM are caused by insufficient insulin action in the target tissues, and pancreatic beta cells are key players in the development of diabetes.

The interaction between adipose tissue and pancreatic beta cells, known as the adipo-insular axis, plays a pivotal role in metabolic regulation and the onset of metabolic disorders [5]. The primary components of the adipo-insular axis include adipokines and insulin. The dynamic relationship between adipokines and insulin is essential for maintaining metabolic homeostasis, and disturbances in this axis can contribute to metabolic dysfunction. The adipo-insular axis represents a bidirectional communication pathway; adipokines such as leptin and adiponectin influence beta cell function and insulin production. For instance, leptin inhibits insulin secretion under normal physiological conditions, whereas adiponectin promotes insulin secretion and beta cell survival [6]. Insulin affects adipocyte metabolism and modulates adipokine secretion. For example, insulin promotes the secretion of leptin, acting as a feedback mechanism to regulate energy intake [7,8]. In metabolic health, the adipo-insular axis ensures a balanced response to changes in energy status. Adipokines and insulin interact dynamically to regulate energy intake, storage, and expenditure, thereby facilitating metabolic adaptation and homeostasis. However, in metabolic disorders, such as obesity and T2DM, the adipo-insular axis is dysregulated. Obesity, which is associated with increased adipose tissue mass, leads to altered adipokine profiles. This profile is characterized by increased levels of leptin, resistin, and TNF-α, and decreased levels of adiponectin. Such alterations in adipokines contribute to insulin resistance and beta cell dysfunction [9,10].

This review aims to investigate the relationship between adipokines and pancreatic beta cells, evaluate their roles in metabolic homeostasis, and elucidate their implications in the pathogenesis of metabolic disorders.

## 2. Beta Cell and Diabetes

### 2.1. Diabetes and Diabetes-Related Complications

Diabetes mellitus is a chronic metabolic disease characterized by elevated blood glucose levels, accompanied by defects in insulin secretion, insulin action, or both. Diabetes is classified into four categories: T1DM, T2DM, gestational diabetes (GDM), and specific types of diabetes due to other causes [11]. T1DM and T2DM are main types of diabetes and are diagnosed using well-established criteria. T2DM is the most common, accounting for more than 90% of all diabetes cases [12]. T1DM arises from the autoimmune destruction of beta cells, while T2DM develops due to a reduced peripheral insulin response, subsequently followed by progressive beta cell dysfunction [3,4]. Both type 1 and type 2 diabetes are influenced by genetic and environmental factors that lead to a loss of β-cell mass and/or function, resulting in hyperglycemia and the associated complications. Chronic hyperglycemia causes damage to macro- and micro-vessels, leading to organ dysfunction and failure such as stroke, cardiovascular disease, kidney failure, nerve damage, and retinopathy.

### 2.2. Glucose-Stimulated Insulin Secretion (GSIS)

The essential role of pancreatic beta cells is to secrete the appropriate amount of insulin into the bloodstream in response to blood glucose levels. To achieve this goal, the beta cell has a specialized glucose-sensing machinery and a vesicle trafficking system [13,14]. Glucose enters beta cells through glucose transporters (GLUTs) and is phosphorylated by glucokinase, a hexokinase with a high Michaelis–Menten constant (Km) and maximal velocity (Vmax) [15]. Since glucose 6-phosphate, the product of hexokinase, cannot cross GLUT, phosphorylated glucose remains trapped in the cytosol, resulting in increased glycolytic flux. The unique properties of glucokinase, specifically its high Km and Vmax values, enable beta cells to sense changes in blood glucose levels between the fasting and postprandial states. Thus, when blood glucose levels are elevated, the glycolytic flux in beta cells increases. This increased glycolytic flux activates the tricarboxylic acid (TCA) cycle and mitochondrial oxidative phosphorylation, leading to increased ATP production. Since beta cells have ATP-sensitive K^+^ (KATP) channels in the plasma membrane that close upon ATP binding, increased ATP production promotes the inhibition of outward K^+^ flux [16]. This results in depolarization of the plasma membrane, followed by the activation of L-type voltage-dependent Ca^2+^ channels and Ca^2+^ influx. The elevation of intracellular Ca^2+^ triggers the fusion of insulin granules with the plasma membrane, leading to increased exocytosis. Several studies have reported that adipokines affect metabolic intermediates, secondary messengers, enzymes, and channels involved in GSIS in beta cells. Consequently, the adipose tissue can communicate with beta cells and modulate the insulin secretion machinery via adipokines, thereby contributing to the maintenance of energy homeostasis.

### 2.3. Beta Cell Compensation and Development of Type 2 Diabetes

T2DM is a multifactorial disease affected by both genetic and environmental factors. Generally, the initial event in the development of T2DM is increased peripheral insulin resistance, which is characterized by a decreased response of the body to insulin. However, not everyone with insulin resistance develops T2DM; concurrent beta cell dysfunction is also necessary [4,17,18]. Insufficient insulin action leads to elevated blood glucose levels, which, in turn, increases the demand for beta cells to secrete more insulin. This results in enhanced insulin secretion and promotes the expansion of the beta cell mass. Initially, this compensatory response maintains the blood glucose at euglycemic levels. However, chronic metabolic workload and stress eventually lead to the failure of beta cell adaptation, resulting in beta cell dysfunction and a reduction in beta cell mass. In this state, termed beta cell failure, blood glucose levels begin to increase, and patients may be diagnosed with diabetes [19,20].

Increased adiposity and ectopic fat storage are major contributors to insulin resistance in peripheral organs such as the liver, adipose tissue, and muscles. Excess nutrients and metabolic stress alter the pathophysiological characteristics of the adipose tissue, leading to changes in adipokine expression and secretion [21,22]. Alterations in adipokine secretion associated with obesity or metabolic disorders affect distant organs including the endocrine pancreas. The secreted adipokines influence the compensatory response of beta cells and contribute to beta cell failure. Therefore, adipokines have been proposed as potential therapeutic targets for the prevention and treatment of diabetes mellitus.

## 3. Leptin

Leptin plays a crucial role in energy homeostasis and body weight control as well as in regulating glucose homeostasis. This was discovered in a study on an obese (ob) mouse strain, a genetic model of obesity [23]. The discovery of leptin and its receptors has provided invaluable insights into the biological mechanisms that regulate body weight and energy homeostasis. This reveals a complex system in which peripheral signals communicate the body’s nutritional status to the brain, which regulates energy intake and expenditure. Leptin functions centrally, primarily by influencing hypothalamic action, and peripherally, by directly interacting with glucose metabolism or by modulating insulin action in tissues such as skeletal muscle, liver, and adipose tissue [24,25,26,27]. Given its comprehensive role in glucose homeostasis, leptin is a critical component in both metabolic health and disease.

The leptin receptor, encoded by the LEPR gene, is a member of the class I cytokine receptor family and exists in several isoforms owing to alternative splicing. The long isoform, often referred to as the functional leptin receptor (Ob-Rb), is the primary isoform that mediates a wide range of biological actions [28]. Beta cells in both rodents and humans express the long isoform of the leptin receptor [29,30,31,32]. Its large intracellular domain is capable of activating multiple signaling pathways, including the JAK2/STAT3, MAPK, and PI3K pathways [33].

Leptin can directly inhibit GSIS in both rodent and human pancreatic beta cells [34,35]. However, the relationship between leptin and insulin secretion appears to be more complex than initially postulated. The effect of leptin on insulin secretion has been suggested to exhibit a U-shaped dose–response curve [32,36]. The exact mechanism by which leptin inhibits insulin secretion is not fully understood; however, several signaling pathways have been implicated. For instance, leptin has been proposed to activate ATP-sensitive potassium (KATP) channels in beta cells, leading to membrane hyperpolarization and suppression of insulin release. Leptin modulates insulin secretion by promoting the translocation of KATP channels to the plasma membrane through signaling pathways, including the AMPK, PKA, and PI3K pathways [37,38,39,40]. Additionally, leptin increases KATP channel trafficking by inhibiting phosphatase and tensin homolog (PTEN) via glycogen synthase kinase 3β (GSK3β) [41]. This process is also affected by the NMDA subtypes of glutamate receptors (NMDARs) and Src kinase-mediated phosphorylation of the GluN2A subunit [42]. In addition, leptin inhibits insulin secretion by decreasing protein phosphatase 1-dependent Ca^2+^ influx [43]. Moreover, leptin upregulates the expression of Rev-erbα, an important gene involved in circadian rhythms, in beta cells [44]. LIM homeodomain transcription factor Isl-1 expression is suppressed by leptin, leading to decreased insulin secretion [45]. Therefore, various signaling pathways contribute to the inhibitory effect of leptin on beta cell insulin secretion.

The protective and harmful effects of leptin on beta cells remain controversial. Leptin prevents fatty acid-induced apoptosis by modulating the expression of the anti-apoptotic gene B-cell leukemia 2 (BCL-2) [46]. Similarly, leptin reduces apoptosis in a beta cell line through an increase in BCL-2 and a decrease in BCL2-associated X protein (BAX) [47]. In contrast, leptin can have deleterious effects on beta cells. High levels of leptin and glucose induce beta cell apoptosis via the JNK pathway [48]. Long-term exposure of human islets to leptin reduces interleukin 1 (IL1) receptor antagonist production by beta cells and causes the release of IL-1β from the islet, resulting in impaired beta cell function, activation of caspase-3, and apoptosis [49]. Leptin-deficient mice showed beta cell apoptosis by inducing the expression of the receptor for advanced glycation end products in beta cells [50].

Furthermore, the role of leptin in insulin secretion and beta cell proliferation has been investigated in tissue-specific leptin receptor-knockout (KO) mice. Beta cell- and hypothalamus-specific KO mice exhibit impaired GSIS and an increased beta cell mass [51]. Similarly, pancreas-specific leptin receptor-KO mice also showed an augmented beta cell mass [52]. These studies suggest that leptin signaling exerts a negative effect on beta cell mass expansion. Additional findings suggested that leptin can alter the cellular redox state of beta cells, potentially acting as a positive regulator of beta cell mass [53]. In addition, leptin affects the expression of lipoprotein lipase in beta cells [54].

Several studies have investigated the effects of hypothalamic leptin and the sympathetic nervous system on beta cell function. One study found that leptin can directly decrease insulin secretion capacity through sympathetic nervous system activation without significantly affecting beta cell mass [55]. Furthermore, the intracerebroventricular infusion of leptin did not affect pancreatic beta cell regeneration in a streptozotocin (STZ)-induced diabetes model [56]. Further investigation revealed that tanycytes, a type of hypothalamic cells, are instrumental in transporting leptin into the brain and regulating metabolism. Selective deletion of the leptin receptor in tanycytes blocks leptin entry into the brain, leading to glucose intolerance because of reduced insulin secretion, potentially through an altered sympathetic nervous system [57].

## 4. Adiponectin

Adiponectin was first reported by four independent research groups between 1995 and 1996, leading to its initial name under various designations, including Acrp30, AdipoQ, apM1, and GBP 28 [58,59,60,61]. Unlike most adipokines, which are elevated in obesity, adiponectin levels are inversely correlated with adiposity [62,63,64]. Adiponectin enhances insulin sensitivity, exhibits anti-inflammatory properties, and is involved in glucose regulation and fatty acid catabolism. A unique feature of adiponectin is its multimeric structure. It circulates in the bloodstream in various isoforms, including trimers, hexamers, and high-molecular-weight (HMW) forms. These multimers have distinct biological activities, and the HMW form is believed to be the most biologically active in terms of insulin sensitization [65]. AdipoR1 and AdipoR2 have been identified as integral membrane proteins with seven transmembrane domains. However, they differ distinctly from classical G protein-coupled receptors (GPCR). Their N-terminal domains are oriented towards the cytoplasm, whereas the C-terminus is extracellular, an orientation opposite to that of most membrane receptors. AdipoR1 is ubiquitously expressed but is predominantly found in skeletal muscle. Conversely, AdipoR2 shows more restricted expression, with the liver being the primary site [66]. The expression of the adiponectin receptors AdipoR1 and AdipoR2 in human and rat beta cells is similar to that in the liver and greater than that in muscle [67]. Adiponectin exerts its actions by binding to the adiponectin receptors AdipoR1 and AdipoR2, leading to the activation of several downstream signaling pathways, including the AMPK, PPARα, and p38 MAPK pathways.

Adiponectin has a complex association with insulin secretion. Several studies have demonstrated a significant influence of adiponectin on insulin secretion [68,69]. Globular adiponectin enhances insulin secretion at high glucose concentrations through AMPK activation [70]. Conversely, globular adiponectin potentiates GSIS through an AMPK-independent pathway, primarily by increasing fatty acid oxidation rather than augmenting glucose oxidation [71]. Additionally, adiponectin stimulated mitochondrial metabolic flux and GSIS in INS1 cells and primary islets [72]. Adiponectin increased PPARγ expression, insulin content, and insulin secretion in MIN6 cells via PPARγ-dependent mechanisms. However, the proliferative effect on these cells was independent of PPARγ activation [73]. Moreover, adiponectin activated both AMPK and acetyl-CoA carboxylase in beta cells. This action inhibits the conversion of glucose-derived carbon into acyl-CoA and cholesterol biosynthetic intermediates [74].

Several studies have suggested that adiponectin protects beta cells from glucotoxicity-induced apoptosis and dysfunction by activating the AMPK signaling pathway [75,76]. Adiponectin activates Akt and ERK, which protects against apoptosis and stimulates insulin gene expression and secretion [77]. Two agonist regions of adiponectin, the globular domain and a small N-terminal region, enhanced cell viability in the rat beta cell line BRIN-BD11, primarily through ERK1/2 activation [78]. Recent studies have reported that adiponectin mitigates islet lipotoxicity and promotes beta cell regeneration in an inducible acute beta cell ablation mouse model [79,80]. Overexpression of adiponectin in STZ mice induces significant anti-diabetic and anti-apoptotic effects by inhibiting the intrinsic and extrinsic apoptotic pathways in beta cells [81]. Similarly, adiponectin protects beta cells against lipoapoptosis by inhibiting the intrinsic apoptosis pathway [82]. Additionally, adiponectin induces low concentrations of reactive oxygen species (ROS) through NADPH oxidase. A physiological increase in ROS levels is associated with enhanced proliferation of beta cells [53].

Adiponectin plays a significant role in the management of insulin resistance and beta cell function, particularly during pregnancy. Alterations in adiponectin levels are associated with the risk and progression of GDM, and may have implications for the future development of T2DM. Weight gain, particularly fat accumulation, is a critical factor in this. Numerous studies have investigated the association between adiponectin levels and beta cell function in GDM. A longitudinal study focusing on Hispanic women with recent GDM revealed that weight gain, particularly fat accumulation, in conjunction with decreased adiponectin and increased C-reactive protein levels was significantly associated with a decline in beta cell function relative to insulin sensitivity [83]. Other studies have similarly highlighted the association of adiponectin with beta cell function and insulin resistance during pregnancy. Lower adiponectin levels during early pregnancy are associated with a higher risk of developing GDM and increased insulin resistance [84]. In late pregnancy, adiponectin concentration was found to be independently associated with beta cell function, suggesting a pivotal role for adiponectin in mediating insulin resistance and beta cell dysfunction, thus contributing to the development of GDM and potentially T2DM [85]. Furthermore, low adiponectin levels during pregnancy predicted postpartum insulin resistance, beta cell dysfunction, and fasting glycemia, especially in women with GDM [86].

Two studies investigated the impact of low adiponectin levels, or hypoadiponectinemia, on GDM and beta cell function during pregnancy using mouse models. The first study reported that adiponectin deficiency led to glucose intolerance, hyperlipidemia, and increased fetal body weight in late pregnancy. These metabolic abnormalities were ameliorated by reintroduction of adiponectin, emphasizing its essential role in managing metabolic adaptations during pregnancy [87]. Another study found that adiponectin deficiency significantly reduced beta cell proliferation and insulin levels. However, the direct manipulation of adiponectin receptors in beta cells did not produce the same effects. Adiponectin was found to promote the expression of placental lactogen (PL), a hormone essential for maternal beta cell proliferation during pregnancy. In adiponectin-deficient mice, PL injection restored beta cell proliferation and insulin levels, indicating a critical role of adiponectin in promoting PL expression and, consequently, beta cell proliferation during pregnancy [88].

## 5. Apelin

Apelin, originally isolated from bovine stomach extracts, was identified as an endogenous ligand for an orphan GPCR, the apelin receptor (APJ) [89]. Apelin is abundantly expressed in the central nervous system (CNS) and peripheral tissues, including the heart, liver, kidney, gastrointestinal tract, and adipose tissues [90,91]. Plasma apelin levels have been found to be increased in both obese and hyperinsulinemic mice and humans [92,93]. Insulin upregulates apelin expression in adipose tissue by activating the PI3K, PKC, and MAPK pathways [92]. Furthermore, APJ is expressed in pancreatic islets and is involved in insulin secretion [94]. Apelin has been found to directly inhibit insulin secretion in rat insulinoma INS-1 cells, particularly in response to glucose and glucagon-like peptide-1 (GLP-1). This process involves the activation of PI3K-dependent phosphodiesterase 3B, which leads to suppression of cAMP levels [95]. Recent evidence suggests that apelin is involved in beta cell proliferation and may offer protection against beta cell mass reduction under diabetic conditions. The apelin–APJ signaling system plays a stimulatory role in pancreatic islet homeostasis and promotes metabolism-induced beta cell hyperplasia. Mice with a pancreas-selective deletion of APJ exhibit a significant decrease in islet size and density, and beta cell mass, resulting in impaired glucose clearance [96]. The administration of Wharton’s jelly-derived mesenchymal stem cells overexpressing apelin effectively lowered blood glucose levels and promoted pancreatic beta cell proliferation in a high-fat diet (HFD)/STZ rat model [97]. In STZ or HFD models, an acylated apelin-13 analog reduced beta-to-alpha cell transdifferentiation, decreased beta cell apoptosis, increased beta cell proliferation, and maintained beta cell identity.

## 6. Resistin

Resistin was initially identified in murine adipose tissue, which sparked significant interest in its potential roles in human metabolism [98]. This initial characterization of resistin led to the hypothesis that it might serve as a pivotal link between obesity and diabetes, largely based on preliminary evidence suggesting that resistin can inhibit insulin activity in rodent models. However, in humans, resistin expression is primarily observed in mononuclear cells and macrophages, with comparatively low expression in adipocytes [99,100,101]. The notable effects of resistin on insulin resistance observed in rodents have not been consistently replicated in human studies, leading to a diminished interest in this molecule among diabetes researchers. Nevertheless, several studies have reported intriguing findings regarding the involvement of resistin in beta cell function. For instance, in beta cell lines, resistin decreases insulin receptor expression [102], and induces beta cell apoptosis [103]. Furthermore, mice expressing resistin exhibited impaired insulin secretory responses to glucose [104]. Another study investigating the long-term effects of central resistin infusion in rats found that resistin increased both first-phase insulin secretion and beta cell proliferation [105].

## 7. Visfatin

Visfatin, also known as nicotinamide phosphoribosyltransferase (NAMPT) or pre-B cell colony-enhancing factor (PBEF), was initially characterized as an insulin-mimetic adipokine [106]. The original study describing its insulin-mimetic action was later retracted due to concerns regarding the reproducibility of its hypoglycemic properties. Despite this, visfatin is still recognized as a significant factor in beta cell function and regulation of glucose homeostasis. Elevated levels of visfatin have been observed in individuals with T2DM and these elevated levels are associated with increased insulin secretion [107]. Mice with NAMPT haploinsufficiency show reduced NAD biosynthesis and impaired GSIS [108]. Similarly, visfatin has been shown to increase insulin secretion and regulate the expression of genes associated with beta cell function in mouse [109]. Furthermore, visfatin has been reported to promote beta cell proliferation and reduce apoptosis through the activation of MAPK- and PI3K-dependent signaling pathways [110]. In contrast, visfatin and its product, nicotinamide mononucleotide (NMN), were found to have no effect on the viability or apoptosis of beta cells but enhanced GSIS during acute exposure [111]. Central administration of visfatin improves glucose homeostasis in diabetic rats by enhancing insulin secretion and beta cell mass [112]. A recent study reported the effects of two distinct forms of extracellular NAMPT on pancreatic beta cell function. Under physiological conditions, extracellular NAMPT existed predominantly as a dimer and preserved the function and identity of beta cells through NAD-dependent mechanisms. Elevated extracellular NAMPT levels, as observed in T2DM, lead to structural and functional changes, notably, an increase in monomeric extracellular NAMPT. This shift was associated with the induction of a diabetic phenotype in pancreatic islets [113].

## 8. Other Adipokines

### 8.1. Adipsin

Adipsin is a serine protease belonging to the trypsin family that plays a pivotal role in the alternative complement pathway [114,115,116,117]. Recent studies have elucidated the metabolic role of adipsin in beta cell function. Adipsin KO mice display glucose intolerance due to reduced insulin secretion, and their isolated islets exhibit reduced GSIS. Reintroduction of adipsin into diabetic mice alleviates hyperglycemia by enhancing insulin secretion. The beneficial effects of adipsin are mediated by the action of C3a on its receptor [118]. Chronic adipsin supplementation in db/db mice ameliorated hyperglycemia, increased insulin levels, and preserved beta cells. The preservation of beta cells occurs by preventing their dedifferentiation and death through the inhibition of DUSP26 [119].

### 8.2. Lipocalin-2

Lipocalin-2 (LCN2), also known as neutrophil gelatinase-associated lipocalin (NGAL), is a member of the lipocalin superfamily. Several studies have reported the expression of LCN2 in adipose tissue [120,121,122]. Under obesity-associated conditions, white adipose tissue exhibits elevated LCN2 levels [123]. LCN2 has been shown to enhance beta cell function in STZ-treated mice and facilitate early adaptive beta cell proliferation in HFD-fed mice [124].

### 8.3. Chemerin

Chemerin, also known as tazarotene-induced gene 2 (TIG2) or retinoic acid receptor responder 2 (RARRES2), is an adipokine that regulates adipogenesis and energy metabolism and plays a role in the regulation of beta cell mass and function [125,126]. Chemerin-deficient mice exhibit glucose intolerance due to impaired GSIS, which is linked to reduced MafA expression in beta cells [127]. Chemerin also promotes beta cell proliferation and improves mitochondrial homeostasis in beta cells [128].

### 8.4. Fibroblast Growth Factor 21 (FGF21)

FGF21, a peptide hormone mainly secreted by the liver, regulates glucose and lipid metabolism as well as insulin sensitivity in various organs, including the liver, pancreas, adipose tissue, and muscle [129,130,131,132,133]. In adipocytes, FGF21 is an adipokine whose secretion is increased by PPARγ and that plays a role in increasing insulin sensitivity in peripheral tissues [133,134,135]. In addition, FGF21 improves beta cell function and survival by activating the ERK1/2 and Akt signaling pathways in pancreatic beta cells [136]. Furthermore, FGF21 has been implicated in pancreatic beta cell regeneration in a mouse model of T2DM [137].

### 8.5. Growth Differentiation Factor 15 (GDF15)

GDF15 is a polypeptide that belongs to the TGF-β superfamily and is expressed in various organs, including the liver, adipose tissue, and kidney [138]. In mouse models, GDF15 has been shown to reduce food intake and body weight. It also alleviates several diseases, such as obesity, diabetes, nonalcoholic fatty liver disease, and cardiovascular disease [139,140,141]. In addition, GDF15 activates the AMPK signaling pathway and reduces fatty acid oxidation and insulin resistance in mice [142]. In beta cells, GDF15 increases GSIS by enhancing the canonical insulin secretion pathway [143]. Moreover, mice with increased GDF15 expression exhibit an increased beta cell mass and decreased apoptosis [144].

### 8.6. Tumor Necrosis Factor-α (TNF-α)

TNF-α is an inflammatory cytokine that increases in the adipose tissue of obese individuals and contributes to insulin resistance [145,146,147]. TNF-α is involved in inhibition of GSIS in pancreatic beta cells and also contributes to apoptosis of these cells [148,149]. Conversely, other studies have suggested that anti-TNF-α therapy improves insulin sensitivity and contributes to the survival of pancreatic beta cells [150,151]. Furthermore, anti-TNF-α therapy delays disease onset in a mouse model of T1DM [152].

## 9. Conclusions

Following the discovery of the first adipokine, leptin, in 1994 [23], numerous studies have been conducted on adipokines. Adipokines have been shown to play crucial roles in regulating energy metabolism, inflammation, insulin sensitivity, and appetite. They exert their effects through complex interactions with pancreatic beta cells, forming the adipo-insular axis. This bidirectional communication pathway ensures metabolic homeostasis under normal conditions, but disruptions in this axis can contribute to the development of metabolic disorders, such as obesity and T2DM.

Pancreatic beta cells secrete insulin to regulate blood glucose levels. In T1DM, autoimmunity destroys beta cells, leading to insulin deficiency, whereas in T2DM, insulin resistance and beta cell dysfunction result in insulin deficiency. Adipokines affect the GSIS machinery of beta cells. For example, leptin directly inhibits GSIS, whereas adiponectin activates GSIS via an AMPK-independent pathway.

In addition, leptin affects insulin secretion through complex mechanisms involving the modulation of potassium channels, intracellular signaling pathways, and the regulation of apoptosis. Although it inhibits insulin secretion under normal physiological conditions, its effects on beta cell survival and proliferation remain controversial. In contrast, adiponectin increases insulin sensitivity and protects beta cells. It stimulates insulin secretion, prevents apoptosis, and promotes beta cell regeneration, particularly during pregnancy.

Apelin, resistin, visfatin, and other adipokines also affect beta cell function and insulin secretion. Apelin influences beta cell homeostasis, and its deficiency can lead to impaired glucose metabolism. The effects of resistin on insulin resistance are not fully understood; however, it appears to influence beta cell apoptosis and insulin receptor expression. Visfatin has been shown to have insulin-mimetic properties and is associated with insulin secretion, beta cell proliferation, and cell viability.

Adipsin, lipocalin-2, chemerin, FGF21, GDF15, and TNF-α also contribute to the complex network of interactions between adipokines and beta cells. These adipokines have been shown to influence beta cell mass, insulin secretion, and overall metabolic regulation. Understanding the intricate relationship between adipokines and pancreatic beta cells provides insights into the development of potential treatment strategies for metabolic disorders such as obesity and diabetes. Figure 1 and Table 1 illustrate the role of adipokines in pancreatic beta cells.

In conclusion, the adipo-insular axis is a critical regulatory system that maintains metabolic homeostasis through the interplay between adipokines and pancreatic beta cells. Dysregulation of this axis may have significant implications in the onset and progression of metabolic disorders, making it a promising area of research for the development of novel therapeutic interventions.

## 10. Methodology

Two authors, J.K. and H.K., independently selected relevant studies from PubMed, MeSH, Scopus, Google Scholar, and Embase. The search strategy incorporated the following keywords or subject headings: adipokine, diabetes mellitus, metabolic disorder, insulin-secreting cells, pancreatic beta cell, insulin, leptin, adiponectin, apelin, resistin, visfatin, adipsin, lipocalin-2, chemerin, FGF21, GDF15, and TNF-α. The search was restricted to studies published in English. The authors compared and reviewed the reference lists for potential relevance. The authors discussed the articles, and 152 papers were considered relevant to the search criteria and suitable for addressing the research objective.

## Figures and Tables

**Figure 1 biomedicines-11-02589-f001:**
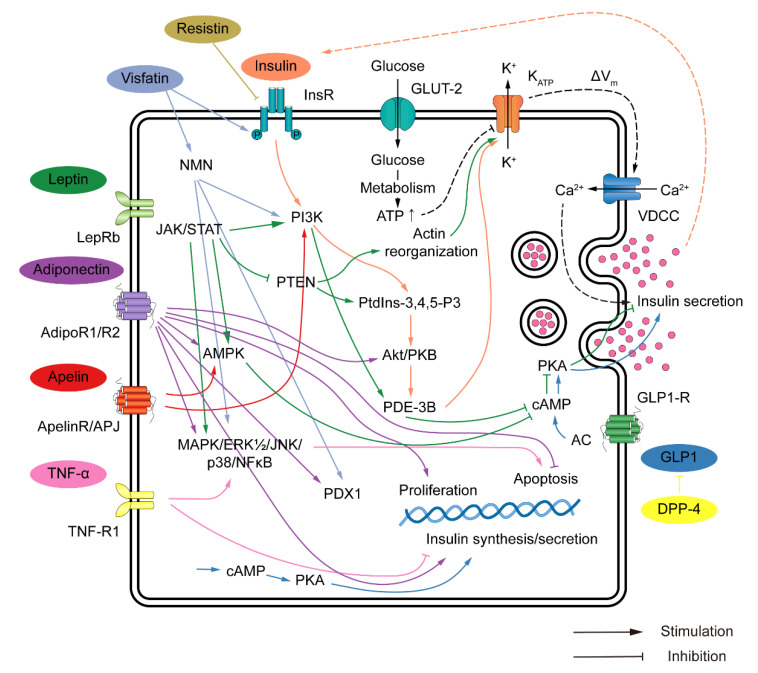
Schematic illustration of the role of adipokines in pancreatic beta cells. AdipoR, adiponectin receptor; Akt/PKB, protein kinase B; AMPK, 5′ AMP-activated protein kinase; ApelinR, apelin receptor; DPP-4, dipeptidylpeptidase-4; ERK, extracellular signal-regulated kinases; GLUT-2, glucose transporter 2; GLP1, glucagon-like peptide-1; InsR, insulin receptor; JAK/STAT, Janus kinase/signal transducer and activator of transcription; JNK, c-Jun N-terminal kinases; KATP, ATP-sensitive potassium channel; LepRb, leptin receptor long isoform; MAPK, mitogen-activated protein kinase; NFκB, nuclear factor kappa-light-chain-enhancer of activated B cells; NMN, nicotinamide mononucleotide; PDE-3B, phosphodiesterase 3B; PDX-1, pancreatic and duodenal homeobox 1; PI3K, phosphatidylinositol 3 kinase; PtdIns-3,4,5-P3, phosphatidylinositol (3,4,5)-trisphosphate; PTEN, phosphatase and tensin homolog; VDCC, voltage-gated calcium channel.

**Table 1 biomedicines-11-02589-t001:** Effects of adipokines on pancreatic beta cells and insulin secretion.

Adipokine	Effect		References
Adiponectin	Insulin secretion	↑	[69,70,71,72,73]
	Beta cell proliferation	↑	[53,88]
	Beta cell apoptosis	↓	[75,76,77,82]
Adipsin	Insulin secretion	↑	[118]
	Beta cell apoptosis	↓	[119]
	Beta cell dedifferentiation	↓	[119]
Apelin	Insulin secretion	↓/↑	[95,97]
	Beta cell proliferation	↑	[97]
Chemerin	Insulin secretion	↑	[127]
	Beta cell proliferation	↑	[128]
FGF21	Insulin secretion	↑	[136]
	Insulin sensitivity	↑	[133]
	Beta cell regeneration	↑	[137]
GDF15	Insulin secretion	↑	[142]
	Insulin resistance	↓	[143,144]
Leptin	Insulin secretion	↓	[34,35,37,38,39,40,43,45]
	Beta cell proliferation	↓	[53]
	Beta cell apoptosis	↓/↑	[46,47,48,49]
Lipocalin-2	Insulin secretion	↑	[124]
	Beta cell proliferation	↑	[124]
Resistin	Insulin secretion	↓	[104]
	Beta cell apoptosis	↑	[103]
TNF-α	Insulin secretion	↓	[148,149]
	Insulin resistance	↑	[145,146,147,150,151]
Visfatin	Insulin secretion	↑	[107,109,111,112]
	Beta cell proliferation	↑	[110]
	Beta cell apoptosis	↓	[110]

## Data Availability

Not applicable.

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
