# Peer review of "The Interplay of Adipokines and Pancreatic Beta Cells in Metabolic Regulation and Diabetes"

_biomedicines, 2023, doi:10.3390/biomedicines11092589_

Round 1

Reviewer 1 Report

Dear Authors,

This is a very interesting study that is related to the Research Topic.

The increased incidence of glycemic imbalances, particularly diabetes, makes it extremely useful to understand the mechanisms involved and, starting from this aspect, to implement targeted therapies.

I have few observations.

Table 1, line 308

For apelin you mentioned "insulin secretion" twice.

Please rephrase.

Kind regards,

Author Response

We deeply appreciated your time and input in reviewing our manuscript. As you recommended, we rephrased in Table 1.

Reviewer 2 Report

Major suggestions:

1. Please add section of METHODOLOGY. What word combinations did the authors use to search for publications that they combined and used when writing the manuscript and in what databases were the searches performed?

2. Please add figure presenting ,,glucose-stimulated insulin secretion''

3. Please add a short chapter focused on diabetes and metabolic disorders

Author Response

1. Please add section of METHODOLOGY. What word combinations did the authors use to search for publications that they combined and used when writing the manuscript and in what databases were the searches performed?

→ We deeply appreciated your time and input in reviewing our manuscript. As you recommended, we add METHODOLOGY section as follows:

  1. Methodology

 Two authors, J.K. and H.K., independently selected relevant studies from PubMed, MeSH, Scopus, Google Scholar, and Embase. The search strategy incorporated the following keywords or subject headings: adipokine, diabetes mellitus, metabolic disorder, insulin-secreting cells, pancreatic beta cell, insulin, leptin, adiponectin, apelin, resistin, visfatin, adipsin, lipocalin-2, chemerin, FGF21, GDF15, and TNF-α. The search was restricted to studies published in English. The authors compared and reviewed the reference lists for potential relevance. The authors discussed the articles, and the 154 papers were considered relevant to the search criteria and suitable for addressing research objective.

2. Please add figure presenting ,,glucose-stimulated insulin secretion''

→ We deeply appreciated your time and input in reviewing our manuscript. As you recommended, we improved our figure 1.

3. Please add a short chapter focused on diabetes and metabolic disorders

→ We deeply appreciated your time and input in reviewing our manuscript. As you recommended, we added as short chapter in result part as follows:

2.1. Diabetes and diabetes-related complications

Diabetes mellitus is a chronic metabolic disease characterized by elevated blood glucose levels, accompanied by defects in insulin secretion, insulin action, or both. Diabetes is classified into four categories: T1DM, T2DM, gestational diabetes (GDM), specific types of diabetes due to other cause [11]. T1DM and T2DM are main types of diabetes and are diagnosed using well-established criteria. T2DM is the most common, accounting for more than 90% of all diabetes cases [12]. T1DM arises from the autoimmune destruction of beta cells, while T2DM develops due to a reduced peripheral insulin response, subsequently followed by progressive beta cell dysfunction [13, 14]. Both type 1 and type 2 diabetes are influenced by genetic and environmental factors that lead to a loss of β-cell mass and/or function, resulting in hyperglycemia and associated complications. Chronic hyperglycemia causes damage to macro- and micro-vessels, leading to organ dysfunction and failure such as stroke, cardiovascular disease, kidney failure, nerve damage and retinopathy.

Round 2

Reviewer 2 Report

Accept in present form